# Assessment of Physical Tests in 6–11 Years Old Children: Findings from the Play Lifestyle and Activity in Youth (PLAY) Study

**DOI:** 10.3390/ijerph20032552

**Published:** 2023-01-31

**Authors:** Dai Sugimoto, Andrea Stracciolini, Laura Berbert, Eric Nohelty, Greggory P. Kobelski, Becky Parmeter, Edie Weller, Avery D. Faigenbaum, Gregory D. Myer

**Affiliations:** 1The Micheli Center for Sports Injury Prevention, Waltham, MA 02453, USA; 2Faculty of Sport Sciences, Waseda University, Tokyo 202-0021, Japan; 3Division of Sports Medicine, Department of Orthopaedics, Boston Children’s Hospital, Boston, MA 02115, USA; 4Harvard Medical School, Boston, MA 02115, USA; 5Division of Emergency Medicine, Department of Medicine, Boston Children’s Hospital, Boston, MA 02115, USA; 6Biostatistics and Research Design Center, Institutional Centers for Clinical and Translational Research, Boston Children’s Hospital, Boston, MA 02115, USA; 7Division of Hematology and Oncology, Boston Children’s Hospital, Boston, MA 02115, USA; 8The College of New Jersey, Ewing, NJ 08628, USA; 9Emory Sports Performance and Research Center (SPARC), Flowery Branch, GA 30542, USA; 10Emory Sports Medicine Center, Atlanta, GA 30329, USA; 11Department of Orthopaedics, Emory University School of Medicine, Atlanta, GA 30307, USA

**Keywords:** exercises, exercise deficits disorder, moderate to vigorous physical activity, physical literacy

## Abstract

The purpose was to evaluate selected physical tests in children and to compare the outcomes by sex. A cross-sectional study design was used to evaluate children 6–11 years who completed five physical tests: hand grip, vertical jump, sit and reach, Y-balance, and obstacle course (time and score). The outcome measures including test results were descriptively examined and compared by sex. The study participants consisted of 133 children (62 males and 71 females, with a median age of 7.8 years). Girls showed superior sit and reach performance (*p* = 0.002) compared with boys. Boys demonstrated better Y-balance scores (*p* = 0.007) and faster obstacle time (*p* = 0.042) than girls. Sex comparison within three age groups (6–<8 years, 8–<10 years, and 10–<12 years) showed that girls performed better on the sit and reach compared with boys in the in 6–<8 years (*p* = 0.009). Boys demonstrated higher Y-balance scores (*p* = 0.017) and faster obstacle time (*p* = 0.007) compared with girls in the 8–<10-year age group. These data will serve to guide future efforts to evaluate normative measures of physical literacy and guide targeted training interventions to promote sustained physical activity in children with deficits relative to their age and sex norms.

## 1. Introduction

Physical inactivity is a global health concern [1,2], especially among children and adolescents [3,4], as it most often sets the trajectory of inadequate motor development [5,6] as well as other undesirable physical developments in adulthood [7,8,9,10]. According to the national guidelines developed by the Centers for Disease Control and Prevention, children and adolescents (age 6–17 years old) should perform an average of 60 min or more of moderate to vigorous physical activity (MVPA) daily [11]. In the guidelines, it is explained that an average of 60 min or more of daily MVPA can enhance cardiorespiratory fitness level, improve body composition, and develop stronger bones and muscles [12]. The additional health benefits of regular physical activity, such as a reduction in cholesterol, fasting blood glucose, and depression in adulthood are reported [13,14,15,16,17,18]. Therefore, establishing regular physical activity habits early in life, ideally a 60-min of MVPA daily as well as high quality movement training [19], is key for future physical, emotional, and psychosocial well-being [12]. One of the recent studies examined how the MVPA is related to health indicators including body composition, aerobic fitness, blood pressure, and quality of life [20]. This study concluded that physical literacy and aerobic fitness were mediated by the MVPA [20]. This study also highlights the substantial impact of MVPA on physical literacy [20].

The term exercise deficit disorder (EDD) was first introduced to the literature almost 10 years ago. EDD is a condition characterized by reduced levels of MVPA [21]. Contributing to EDD, physical illiteracy is a term often used in a discussion of a lack of movement competence, especially among children [22,23,24]. The Canadian Assessment of Physical Literacy (CAPL) defined physical literacy as motivation, confidence, physical competence, knowledge, and understanding to value and take responsibility for engagement in physical activities for life [25]. Two other reports postulated that physical literacy consists of four components: motivation and confidence, daily behavior, physical competence, and knowledge and understanding [26,27]. In the evaluation of physical literacy, a positive association was found between high physical literacy and greater cardiovascular fitness in 8–12 years old children [28]. Among the four sub-components, the effective assessment of physical competence is a topic of debate [29]. There are a variety of procedures including physical activity level by questionnaires [30,31,32,33], fundamental movement skills test [29], and other physical tests including the Bruininks–Oseretsky test [34], agility [35], and multi component fitness tests that can support evaluations of physical competence in youth [36].

One of recent reports synthesized how physical literacy, physical activity, and health consequences are inter-related [37]. In this article, physical literacy was conceptualized with motor competence/proficiency, predilection (motivation), enjoyment of physical education, and perceived competence [37]. Among them, deficits and disorders in motor coordination were discussed as a factor to hinder health status [37]. Additionally, according to a recent systematic review, physical tests are feasible to measure physical literacy [29]; however, the reliable quantification of physical literacy is currently not well defined [29]. Therefore, it is suggested to investigate better methods for reliable quantification to improve strategies to overcome EDD in youth [29]. Furthermore, sex differences in movement competence during early childhood are not well understood. Understanding potential sex differences in physical measures has important clinical implications in the development of future interventions targeting EDD based on sex in youth. Therefore, the purpose of this study was to evaluate selected physical tests in children (age 6–11 years old) and to compare the outcomes by sex.

## 2. Materials and Methods

### 2.1. Study Design and Participants

The current study is a part of the Play, Lifestyle and Activity in Youth (PLAY) study which consists of a questionnaire that was designed by the team of investigators to examine physical literacy [38] and selected physical tests in children. Physical test data were analyzed cross-sectionally. Study crews visited local YMCAs within suburban Boston, MA, USA and received an approval from YMCAs to recruit participants. Parents or legal guardians of children ages 6–11 years were consented when picking up their children from the YMCA afterschool program. Written or verbal assent was obtained at the same time from participating children. Inclusion criteria were (1) children who were 6–11 years of age and (2) children who were able to perform a series of physical tests. Exclusion criterion was children who physically could not perform physical tests due to chronic diseases or physical handicaps. All study participants received reimbursement for their time. The study was performed before the COVID-19 pandemic (February 2018–August 2019). IRB approval was received from the study institution prior to commencement of the PLAY study.

### 2.2. Physical Tests

In the PLAY study, four physical tests were used to assess specific physical abilities: (1) strength by hand grip, (2) power by vertical jump, (3) flexibility by sit and reach, and (4) balance by Y-balance. An obstacle course was designed to evaluate function and coordination of lower extremity, upper extremity, and entire body movement through hopping, kicking, catching, throwing, and sliding skills. Details of each test and rationale are described below.

#### 2.2.1. Strength: Hand Grip Test

Muscular strength is generated by the contraction of muscles. There are several techniques quantifying strength including isometric, isokinetic, and isotonic methods [39,40,41]. Furthermore, muscular strength can be measured from various procedures with different body parts such as the leg press [42,43], arm curl [44,45], and trunk function [46,47]. Among different strength testing methods, the hand grip test may be the most feasible test. One study examined hand grip strength reliability among 7–12 years old children and reported excellent intra- and inter-rater reliability with intraclass correlation coefficients (ICC) of 0.95 and 0.98 [48]. Another study that focused on leg strength reliability among children showed lower intra- and inter-rater reliability values (ICC of intra-rater reliability: 0.62; ICC of inter-rater reliability: 0.91) [49]. Furthermore, inferior reliability of upper extremity strength measurements including shoulder abduction [50], shoulder external rotation [50], and protraction [51] were documented. The low reliability of shoulder strength may stem from the complexity of the testing position. Regarding the level of complexity, a hand grip test is not too difficult to perform [52,53,54]. Children sit on a seat with both feet touching the ground, and a grip strength dynamometer is placed anteriorly with approximately 90 degrees of elbow flexion. Then, the tester instructs the participant to give a maximum effort for 3 s, three attempts per hand.

#### 2.2.2. Power: Vertical Jump Test

Vertical jump test was chosen to assess power for several reasons. First, similar to the hand grip test, instruction was simple compared with other power assessment tests such as the explosive movements including power clean [55,56], power snatch, [57,58], and power jerk [59,60] as well as the other maneuvers such as leg press [61], back squat [62,63], and bench press [64]. In addition to the complex movements of these tests, power assessment tests require qualified professionals who have experience performing this type of assessment. In order to assess explosiveness efficiently with minimal equipment and space, we thought the vertical jump test was the most appropriate for children and adolescents. Among a few types of vertical jump tests including the drop vertical jump [65,66], submaximal vertical jump [67,68], squat vertical jump [69,70,71], and countermovement vertical jump [72,73,74] with arm swing may be the most adaptable for children and adolescence. Participants were allowed to use arm movements, but no running into or taking extra steps prior to the jump was permitted. Along with swinging movements of the arms, they flexed their knees and hips while lowering their trunk, and then were instructed to jump as high as possible touching the wall to mark peak jump performance.

#### 2.2.3. Flexibility: Sit and Reach Test

To quantify flexibility, sit and reach was selected because it is frequently used among children and adolescents [75,76]. Furthermore, several studies that assessed physical literacy skills employed the sit and reach test to determine flexibility [77,78]. According to a study that synthesized a total of 34 independent studies, sit and reach was found as the most accurate test to determine flexibility through hamstring muscle extensibility [79]. Moreover, one study examined whether body shape status such as obesity alters the sit and reach outcome, but this study concluded that there is no effect of body shape on the sit and reach results [80]. Instruction of this test is also clear, and it is safe for children and adolescents to perform. They are asked to sit down with their knees straight and both feet placed against the sit and reach box. Then, they were asked to bend forward and reach towards their toes as far as they can while their hands on top of each other; the distance from their heel to tip of fingers is recorded.

#### 2.2.4. Balance: Y-Balance Test

There are a few different balance tests such as single-limb standing balance [81,82,83], balance error scoring system [84,85,86], and pediatric balance scale [87,88]. Some of those tests such as balance error scoring system are designed to assess a clinical pathology such as traumatic brain injury [89,90,91]. Furthermore, those tests were performed statically in nature, meaning that standing either single or double legs with maintaining balance in different conditions such as standing with eyes closed [84,86]. Since we test healthy youth and also aim to evaluate ability to control postural stability under dynamic movements, we considered the Y-balance as the most appropriate. Y-balance was documented as a feasible and reproducible evaluation method of dynamic balance control in children and adolescents [92,93]. Additionally, Y-balance was suggested as a part of field-based assessments to examine neuromuscular control [94]. They were instructed to stand with a single leg first. From this position, they were asked to use their toes of the other leg to reach out anterior, posteromedial, and posterolateral directions while maintaining a balance. During this test, they were asked to keep their hands on their waist and to keep the heel of their standing leg on the ground.

#### 2.2.5. Coordination: Obstacle Course

The obstacle course was developed based on the Canadian Assessment of Physical Literacy (CAPL) [27,95], which was designed to assess physical literacy of children and adolescents using a holistic approach and consisted of 4 components: (1) knowledge and understanding, (2) motivation and confidence, (3) daily behavior, and (4) physical competence [27]. In the physical competence domain of the CAPL, obstacle course was illustrated [95]. A major goal of the obstacle course is to assess coordination associated with physical literacy in both lower and upper extremity tasks such as hopping with single and double legs, sliding to the side, kicking a ball to a target, and catching and throwing a ball. An innovative part of the obstacle course is to give an opportunity to evaluate physical literacy of children and adolescents both quantitative and qualitative perspectives. For the quantitative measures, total time spent to complete the obstacle course was measured. For qualitative measures, the following criteria were used:

#### 2.2.6. Lower Extremity Unilateral and Bilateral Function and Coordination: Hop Test

In this test, several objects were placed in a row. Then, children and adolescents were instructed to hop over objects. Two different stations were developed: single leg hop (unilateral hop) and double leg hop (bilateral hop) stations. In the single leg hop, children and adolescents were asked to hop on three round objects three times consecutively. They were instructed using the same leg three times and aim to locate their leg in center of each circle. The objective of this test was to evaluate motor function of unilateral limb. A rater evaluated whether their leg was placed in center of the circle or not. Furthermore, if they fell down during this test, points were deducted. For the double leg hopping, children and adolescents jump over a 10-cm high obstacle three consecutive times. A rater inspects whether they are landing with both feet together at the same time. Moreover, status of a transition from landing to subsequent jumping needs to be carefully checked to ensure there is no pause between each set of landing and jumping. If an extra hop was observed, points were deducted. Furthermore, if contact was made to the obstacles during the jump, rater recorded it and again points were deducted. In the double leg hop, how to attenuate forces using both legs and the ability to transfer the force to subsequent hop needs to be closely assessed. Those careful inspections led to qualitative assessment of hop maneuvers.

#### 2.2.7. Lower Extremity Coordination through Generating Force: Kicking a Ball Test

In this task, children and adolescents were instructed to kick a soccer ball straight to targeted area. The targeted area is 4.3 m away straight from a kicking area, and 1 m wide, designed by cones. In addition to the 1.0 m wide target zone, soccer cones were added 0.5 m away. A rater examines their approach to the kicking maneuver with a continuous, smooth motion. Furthermore, whether or not a non-kicking leg is deliberately planted at the time of kicking needs to be carefully examined. If either the smooth approach to kick a soccer ball or elongated stride pattern to provide an impact to kick a ball is observed, points were given. Finally, a rater checks if the soccer ball hits within the 1.0 m of targeted area or the sub-divided area (0.5 m apart from each cone).

#### 2.2.8. Upper Extremity Function with Eye-Hand Coordination: Catching a Ball Test

Aim of this test is to evaluate upper extremity function through catching a ball. Catching a ball requires eye–hand coordination. Goal of this test was to examine if children and adolescents can sufficiently demonstrate an ability to catch a ball. Eye–hand coordination is known as a key component, which is essential to perform most sporting activities [96]. The ball is tossed by a thrower who is standing 4.2 m away at a 45-degree angle from the participant. A ball is gently tossed from a three meter anteriorly and 45 degrees diagonally. Quality of catching such as catching a ball without fumbling, juggling, or dropping is evaluated by a rater. Catching a ball securely without those problems was perceived as good upper extremity function, and there is no deduced point. However, if a ball is fumbled or juggled, points are deducted.

#### 2.2.9. Upper Extremity Coordination through Generating Force: Throwing a Ball Test

Upper extremity coordination is measured by throwing a ball towards a target. Children and adolescents throw a ball to a target, which is located to 4.0 m away. A sound throwing movement is evaluated in this test. The sound throwing movement included (1) a forward step is opposite (contralateral) to throwing arm, (2) throwing arm comes behind, especially hand goes over shoulder, (3) transferring body weight with adequate body rotation is observed, and (4) arms and shoulders follow the path of the ball release. Finally, an overall body balance is maintained during a throwing movement is checked for qualitative measures. In addition to the throwing mechanics changes, whether the thrown ball hit the center of the target or not is also assessed. If the ball hits outside of the targeted zone, points are deducted.

#### 2.2.10. Overall Function and Coordination including Lower and Upper Extremities: Sliding Test

In this test, children and adolescents were asked to transfer tennis balls from one cone to another with maximum speed. The cones are placed in a zig-zag pattern, 2.0 m apart and 1.0 m diagonally in front. Thus, in order to transfer a tennis ball from one cone to another, they can either slide, shuffling their legs to the side to maintain athletic position or running to the station without shuffling their legs or having athletic position. To differentiate sliding and non-sliding movements, rater checks body alignment. Body alignment such as feet, hips, and shoulders are squared to targeted direction, the movement was rated as non-sliding. Additionally, whether they maintain athletic position or not is based on maintaining a low center of gravity with sufficient knee flexion. For upper extremity coordination, the ability to pick and place a ball on each cone was recorded.

### 2.3. Main Outcome Measures

Scores obtained from hand grip, vertical jump, sit and reach, Y-balance, and obstacle course tests (time and points) were used for main outcome measures. The grip strength score is the summation of the left-side average and the right-side average, each of which is the average of three trials. The vertical jump and sit and reach scores are the average of three trials. The Y-balance score was the average of left and right composite scores, which were calculated by a sum of the three reach directions (anterior, posteromedial, and posterolateral) divided by three times the limb length and multiplied by 100 [97]. The obstacle course diagram and point system are shown in Appendix A

### 2.4. Statistical Analysis

Descriptive statistics reported were count and percentage (%) for categorical variables such as sex, race, and grade while median, interquartile range (IQR) and range (minimum, maximum) were used for continuous variables including age, physical characteristics, and physical tests. Wilcoxon rank sum test was used to compare each of the six outcome measures by sex overall and three age groups: 6–<8 years, 8–<10 years, and 10–<12 years. *p*-values less than 0.05 were considered statistically significant. All analyses were conducted using R software, version 4.0.2 (R Project for Statistical Computing, Vienna, Austria).

## 3. Results

### 3.1. Participants Demographics

A total of 133 children (62 males and 71 females) were enrolled in physical assessment tests. The majority of participants were first, second, and third graders, and the median age was approximately seven and half years (Table 1). Over half of this group identified themselves as Caucasian/white race/ethnic background (Table 1). Other demographics including physical characteristics were found in Table 1. Sex distributions were about equal (Table 1), and there were no significant differences in physical characteristics by sex [Age (years): males; median 7.6, IQR (6.6, 8.7), range 6.0, 11.5 vs. females; median 7.9, IQR (6.9, 9.4), range 6.0, 11.9, *p* = 0.153; Height (cm): males; median 126.5, IQR (122.5, 137.5), range 110.0, 150.0 vs. females; median 129.5, IQR (123.5, 140.5), range 105.0, 162.0, *p* = 0.435; Weight (kg): males; median 26.3, IQR (23.0, 34.0), range 18.1, 58.0 vs. females; median 29.0, IQR (22.7, 36.5), range 17.2, 95.3, *p* = 0.226); BMI: males; median 16.2, IQR (15.0, 17.5), range 12.4, 27.2 vs. females; median 17.0, IQR (15.6, 19.9), range 13.2, 40.7, *p* = 0.061].

### 3.2. Physical Activity Levels

Physical activity data were available for 67.6% of participants (90/133). According to participants, 71.1% (64/90) believed that they perform at least 60 min of MVPA every day. Since participants were young, the MVPA level was questioned to the parents or legal guardians of each participant. According to their feedback, 84.4% (76/90) of the participants spend a minimum of 60 min of MVPA on a daily basis. Furthermore, parents or legal guardians of the participants responded that the daily physical activity time of the participants on weekdays and at the weekend is 2 h and 3 h (both median values).

### 3.3. Physical Assessment and Comparisons by Sex

Table 2 presents the results of the six main outcome measures, and there were significant sex differences in several tests. Girls showed superior sit and reach performance compared with boys (Table 3). Conversely, boys demonstrated better Y-balance scores (Table 3) and faster obstacle times than girls (Table 3). Figure 1, Figure 2, Figure 3, Figure 4, Figure 5 and Figure 6 (horizonal line: median, lower, and upper part of the box: 1st and 3rd quartile, vertical lines from box: 1.5 x quartile, circles: outliers) shows sex comparisons within three age groups (6–<8 years, 8–<10 years, and 10–<12 years). In summary, girls performed better on sit and reach than boys in 6–<8 years group (Figure 4). Boys demonstrated higher Y-balance scores (Figure 2) and faster obstacle time compared with girls in 8–<10 years group (Figure 5). There were no differences in the six outcome measures in the 10–<12 years age group.

## 4. Discussion

The purpose of the current study was to assess selected physical tests in children (age 6–11 years old) and to compare the physical test results by sex. The salient finding of the current study was that sex differences were shown on several physical tests including flexibility (sit and reach, Figure 4), balance (Y-balance, Figure 5), and coordination (obstacle course time, Figure 6) in early childhood (6–11 years). Significantly better flexibility was observed in girls compared with boys (Table 3), and statistically different flexibility was found in 6–<8 years group with 7.5 cm median value differences (Figure 4). It is interesting to note that the flexibility differences were observed as early as 6–<8 years group (Figure 4). Furthermore, according to our data, the flexibility differences keep diverging. The superior flexibility of girls was also noted in 8–<10 and 10–<12 years groups with median value differences of 8.5 cm and 23.7 cm (Figure 4). Our finding was supported by past reports that used sit and reach test [98,99,100]. Superior flexibility were reported in 6–12 years old girls compared with their same age male counterparts [98]. Another study investigating 7–14 year old children reported better flexibility in girls than boys. [100] One study examined the flexibility of 5–6 year old children, and according to this study, the flexibility differences by sex were subtle in this age group [99]. Synthesizing those reports and our findings, flexibility alteration may begin during the ages of 6–<8 years.

In the balance test, boys performed better than girls (Table 3), and when it was broken down by the three age groups, the significant differences were found in the 8–<10 year category with the median composite score value differences of 12.1 (Figure 2). In our search, no studies compared Y-balance by sex in early childhood (6–11 years); however, there is one study that examined the feasibility and reliability of the Y-balance test in this age group [92]. This study identified moderate-to-good levels in the reliability assessment with less than 10% of typical errors in this population [90]. It was reported that maturation of basic motor skills usually occurs in ages between 5–8 years [101]. In our data, variabilities such as the wide range of IQR, min, and max were found in age 6–<8 years, but these values become narrower in 8–<10 and 10–<12 age groups (Figure 2), which may be reflective of the process of basic motor skill development in early childhood. Regarding the sex differences, one review study synthesized 57 Y-balance related investigations and concluded that Y-balance is greatly influenced by age, sex, and sport [102]. Thus, more studies are warranted to encompass effects of age and sex on balance in early childhood.

About coordination, overall, boys completed the obstacle course faster than girls (Table 3). Although boys demonstrated faster obstacle course times in all three age groups, a significant difference was identified in the 8–<10 year group (Figure 5). The obstacle course was designed to assess an overall coordination in both lower and upper extremities, and we measured both quantitative (time) and qualitative (points) measures. It is intriguing to note that obstacle points were comparable (Figure 6), but the obstacle times were different by sex (Figure 5, Table 3). The difference may stem from how boys and girls are motivated to complete the obstacle course. In our observation, some boys were a little more competitive and ambitious to finish the course in a shorter time. Unlike boys, girls listened to our instruction more carefully and showed willingness to perform the assigned tasks neatly. The readiness to perform each task well in girls was not shown in the data scores based on the pre-defined point system (Appendix A), but from observational standpoints, their mindset played a major role completing the obstacle course. In addition to the potential mindset differences between boys and girls, a few studies pointed out a lower physical activity level including free play in girls than boys in childhood [103,104,105]. This may be an actual reason why the obstacle course times were slower in girls compared with boys. To promote more physically active lifestyles, not only in girls, but males as well, one study suggested three practical tips including the use of afternoon time, the enjoyment of PE class, and the support from family members [102].

Several limitations need to be stated. First, although we had 133 signed participants, we could not test all participants or obtain the necessary information fully including physical activity levels. This was mainly due to time constraints. The data collection was performed during YMCA sessions. While we were testing or collecting demographic data, sometimes parents/legal guardians came early to pick up their sons and daughters. In most cases, they left in the middle of the data collection. For instance, race/ethnic background of over a quarter of our participants was missing (Table 1). Although Table 1 showed more than a half of the participants were Caucasian/White, in our impression, the children who participated were a lot more diverse. However, it is difficult to verify race/ethnic information. Second, the sample size was relatively small compared with other physical literacy-related studies. Additionally, our participants were recruited from the YMCA, which may not be a representative group of the general pediatric population. Those two aspects pose a question in generalizability. Third, in our physical tests, some children showed up wearing a pair of non-athletic shoes such as a pair of sandals and rain boots, which might have impacted the results. We should have given more detailed, robust recommendations of bringing his/her own pair of athletic shoes for our testing. Although some children did not wear athletic shoes, there were no injuries during our physical assessment. Lastly, the current study was a cross-sectional study design. Longitudinal study design needs to be employed if the long-term effects in conjunction with the current physical assessment data is sought. Lastly, the current study was a cross-sectional study design and performed prior to the COVID-19 pandemic. It would be intriguing if we were capable of repeating the same set of physical tests to examine the potential effects of COVID-19 in conjunction with physical activity in the pediatric population. To make the current findings more feasible, longitudinal study design needs to be employed to determine the long-term effects.

## 5. Conclusions

The current study did not find sex differences in hand grip, vertical jump, and obstacle course points; however, there were differences in sit and reach, Y-balance, and obstacle course time between boys and girls in early childhood. Future studies are warranted to investigate how those physical test parameters in early childhood impact physical activity participation, health, and other well-being in adolescence and adulthood.

## Figures and Tables

**Figure 1 ijerph-20-02552-f001:**
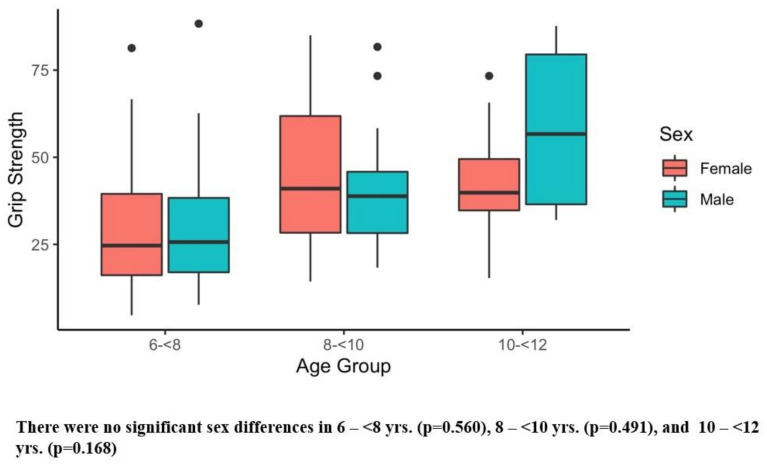
Hand Grip. Black dots are outliers.

**Figure 2 ijerph-20-02552-f002:**
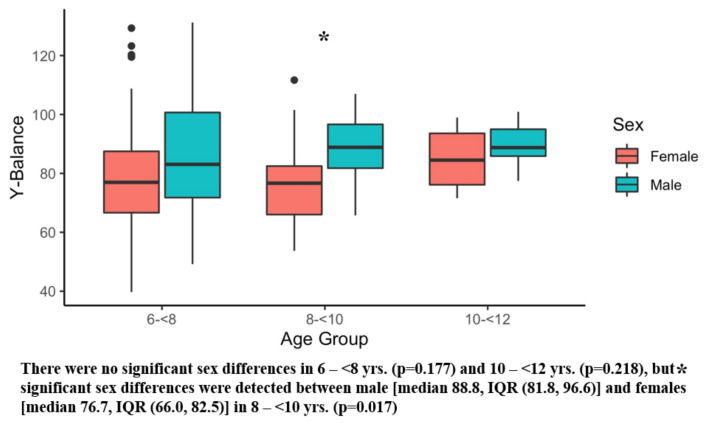
Y-balance. Black dots are outliers.

**Figure 3 ijerph-20-02552-f003:**
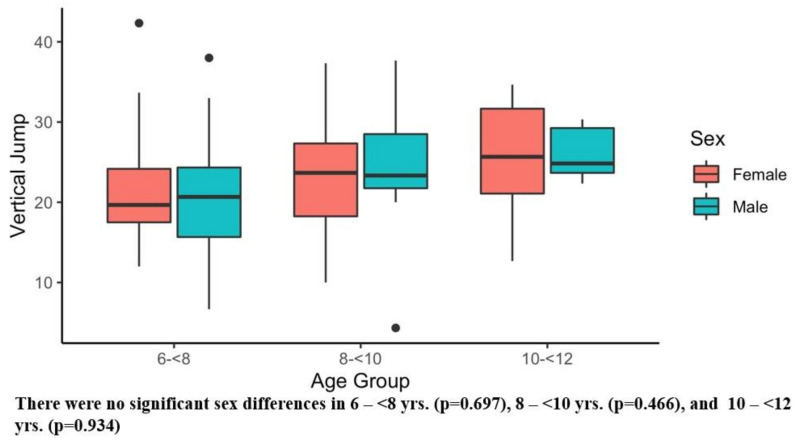
Vertical Jump. Black dots are outliers.

**Figure 4 ijerph-20-02552-f004:**
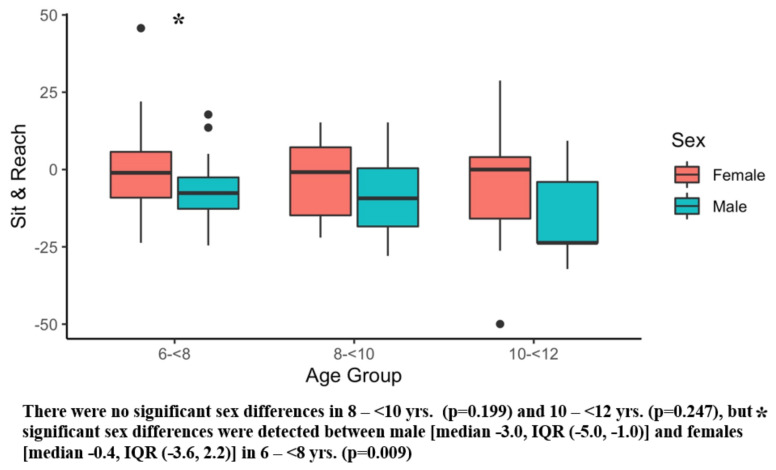
Sit and Reach. Black dots are outliers.

**Figure 5 ijerph-20-02552-f005:**
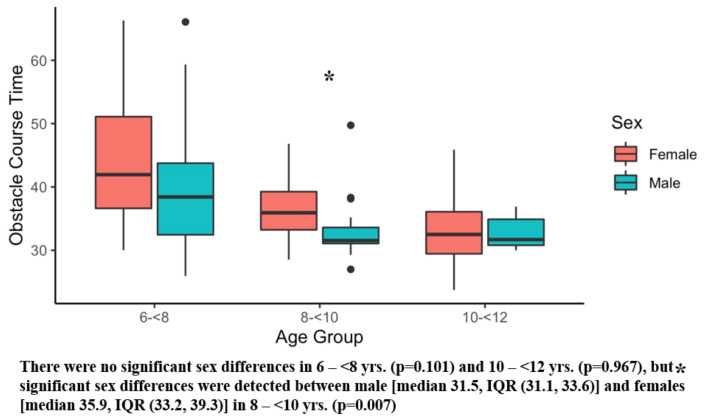
Obstacle Course Time. Black dots are outliers.

**Figure 6 ijerph-20-02552-f006:**
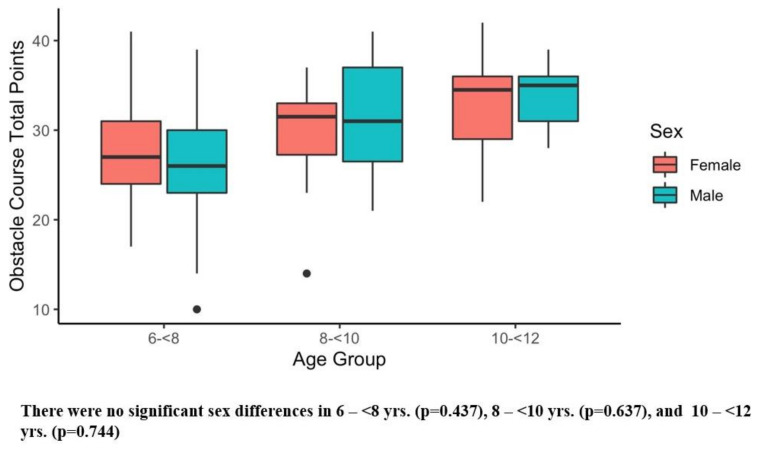
Obstacle Course Points. Black dots are outliers.

**Table 1 ijerph-20-02552-t001:** Participants Demographics.

Variables		Missing	N	Count (%) or Median (IQR) [Min, Max]
**Sex**		0 (0.0%)	133	
	Male Female			62 (46.6%) 71 (53.4%)
**Race/Ethnic Background**		45 (33.8%)	88	
	African American/Black Asian/Pacific Islander Caucasian/White Other Unknown			5 (5.7%) 4 (4.5%) 49 (55.7%) 5 (5.7%) 25 (28.4%)
**Grades**		58 (43.6%)	75	
	Kindergarten First Second Third Fourth Fifth Sixth Seventh			6 (8%) 18 (24%) 20 (26.7%) 14 (18.7%) 8 (10.7%) 7 (9.3%) 1 (1.3%) 1 (1.3%)
**Physical Characteristics**				
Age		0 (0.0%)	133	7.8 (6.8, 9.3) [6.0, 11.9]
Height (cm)		7 (5.3%)	126	128.5 (123.0, 138.0) [105, 162]
Weight (kg)		13 (9.8%)	120	27.2 (22.9, 35.8) [17.2, 95.3]
BMI		15 (11.3%)	118	16.4 (15.2, 18.6) [12.4, 40.7]
Handedness		13 (9.8%)	120	
	Right Left			108 (90.0%) 12 (10.0%)

**Table 2 ijerph-20-02552-t002:** Descriptions of Physical Assessment.

Tests	Missing	N	Median (IQR) [Min, Max]
Hand Grip (kg)	5 (3.8%)	128	34.0 (20.9, 46.7) [4.7, 88.3]
Vertical Jump (cm)	10 (7.5%)	123	22.3 (18.0, 26.2) [4.3, 42.3]
Sit and Reach (cm)	7 (5.3%)	126	−4.2 (−14.4, 2.0) [−19.7, 18.0]
Y Balance (composite score)	6 (4.5%)	127	165.3 (72.1, 93.9) [79.4, 262.4]
Obstacle Course Time (seconds)	10 (7.5%)	123	36.3 (32.4, 43.3) [23.7, 66.3]
Obstacle Course Points (max of 47 pts)	11 (8.3%)	122	29.0 (24.0, 33.0) [10.0, 42.0]

**Table 3 ijerph-20-02552-t003:** Comparisons of Physical Assessment between Males and Females.

Tests	Missing	N	Males (IQR) [Min, Max]	Females (IQR) [Min, Max]	*p*-Values
Hand Grip (kg)	5 (3.8%)	128	33.3 (22.1, 47.1) [7.7, 88.3]	36.8 (19.3, 46.7) [4.7, 85.0]	0.784
Vertical Jump (cm)	10 (7.5%)	123	22.7 (18.7, 26) [4.3, 38.0]	21.7 (18.0, 26.3) [10.0, 42.3]	0.881
Sit and Reach (cm)	7 (5.3%)	126	−3.3 (−6.8, −0.8) [−12.7, 7.0]	−0.3 (−4.8, 2.5) [−19.7, 18.0]	0.002 *
Y Balance (composite score)	6 (4.5%)	127	87.7 (77.3, 97.2) [49.2, 131.2]	77.3 (71.4, 88.1) [39.7, 129.3]	0.007 *
Obstacle Course Time (seconds)	10 (7.5%)	123	35.1 (31.2, 41.1) [25.9, 66.0]	37.3 (33.5, 44.3) [23.7, 66.3]	0.042 *
Obstacle Course Points (max of 47 pts)	11 (8.3%)	122	29.0 (24.0, 32.0) [10.0, 41.0]	29.0 (25.0, 34.0) [14.0, 42.0]	0.424

* *p* < 0.05.

## Data Availability

Most of the relevant data were presented in the manuscript. The data are not publicly available due to the privacy of participants.

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
