# Peer review of "Assessment of Physical Tests in 6–11 Years Old Children: Findings from the Play Lifestyle and Activity in Youth (PLAY) Study"

_ijerph, 2023, doi:10.3390/ijerph20032552_

Round 1

Reviewer 1 Report

The article 'Assessment of Physical Literacy in 6-11 year old Children: Findings from the Play Lifestyle & activity in Youth (PLAY) Study' by Dai Sugimoto et. al. presented to me for evaluation is a valuable item for publication in the International Journal of Environmental Research and Public Health.

The abstract is written in a correct and clear manner.

The introduction adequately introduces the reader to the following subsections, justifies the purpose of the study undertaken and the research gap. 

The material and methods chapter is described sufficiently and correctly. 

The results are presented in 4 tables and 6 figures, most of the results should be considered interesting, but some remain quite predictable and obvious. A graphic change of the figures and their oversized description could be considered. 

The discussion is written correctly with reference to the relevant literature. 

The conclusion is blandly repetitive of the purpose, it would ask you to rewrite and capture the essence of your research. 

The literature needs to be rewritten according to ACS Style, it is worth considering whether it needs to be so extensive, after all, this is not a review, and stale publications can also be considered. 

However, the study has some limitations, we cannot assume that the sample is representative, so the results cannot be generalized. In addition, the study was conducted before the COVID-19 pandemic so we don't know what negative effects the pandemic had, in addition, it would be arch-interesting to study children after the pandemic and thus guide further research. 

Nevertheless, after minor revisions, I accept the article for publication in IJERPH.

Greetings!

Author Response

General Comments: The article 'Assessment of Physical Literacy in 6-11 year old Children: Findings from the Play Lifestyle & activity in Youth (PLAY) Study' by Dai Sugimoto et. al. presented to me for evaluation is a valuable item for publication in the International Journal of Environmental Research and Public Health.

Authors’ Response:  We greatly appreciate your positive comments.

Comment 1. Abstract: The abstract is written in a correct and clear manner.

Response 1: Thank you for your positive comments.

Comment 2. Introduction: The introduction adequately introduces the reader to the following subsections, justifies the purpose of the study undertaken and the research gap. 

Response 2: Thank you for your positive comments.

Comment 3. Methods: The material and methods chapter is described sufficiently and correctly. 

Response 3: Thank you for your positive comments.

Comment 4. Results: The results are presented in 4 tables and 6 figures, most of the results should be considered interesting, but some remain quite predictable and obvious. A graphic change of the figures and their oversized description could be considered. 

Response 4: Thank you for your recommendation. Based on your suggestion, we eliminated table 2.

Comment 5. Discussion: The discussion is written correctly with reference to the relevant literature. 

Response 5: Thank you for your positive comments.

Comment 6. Conclusion: The conclusion is blandly repetitive of the purpose, it would ask you to rewrite and capture the essence of your research. 

Response 6: Thank you for your suggestion. We made this section concise and aimed to send a straight-forward message to readers.

Comment 7. Others: The literature needs to be rewritten according to ACS Style, it is worth considering whether it needs to be so extensive, after all, this is not a review, and stale publications can also be considered. 

Response 7: Thank you for your input. We corrected our citation style for the IJERPH with switched the order of citations and periods in the whole manuscript.  Additionally, based on your recommendation, we aimed to eliminate stale publications from the manuscript. 

Comment 8. Others: However, the study has some limitations, we cannot assume that the sample is representative, so the results cannot be generalized. In addition, the study was conducted before the COVID-19 pandemic so we don't know what negative effects the pandemic had, in addition, it would be arch-interesting to study children after the pandemic and thus guide further research. 

Response 8: Thank you for your comments. We added your input in our limitation section including the potential COVID-19 impact.

Comment 9. Final Comments: Nevertheless, after minor revisions, I accept the article for publication in IJERPH.

Response 9: We are greatly appreciative of your final comment.

Reviewer 2 Report

The purpose of the study was to evaluate tests of physical literacy and to examine sex differences.  I couldn’t see any evidence of test evaluation in this manuscript.  I also couldn’t see any evidence of controlling for chance findings that are likely to occur with any categorisation – in this case sex (And why not gender? It wasn’t clear how this information was obtained).  The participants were recruited from YMCAs and the sample size was quite small, so it is difficult to claim these participants are representative.  I can see this is part of a larger study and perhaps more data from the larger study need to be included to make this manuscript more meaningful.

The design flaws and the overall lack of significance of the research is difficult to overcome with revisions.  In case revisions are requested, I have included the following comments.

Abstract – Please make sure it complies with MDPI requirements of 200 words or less – see 2.4 https://www.mdpi.com/authors/layout

Full stop/period should be after the citation, not before.

The first paragraph has a focus on physical activity (PA) rather than physical literacy (PL) and there is reference to exercise deficit disorder (EDD) throughout the introduction without establishing a clear connection between EDD and PL – I also don’t think it is necessary to make this case. That said, if the authors want to make the case that PL is important because it could lead to strategies to overcome EDD (line 76), then they need to explain how PL relates to PA (and how PA relates to EDD).  These are not direct causal relationships.  Please also see Caldwell, Hilary AT, et al. "Physical literacy, physical activity, and health indicators in school-age children." International Journal of Environmental Research and Public Health 17.15 (2020): 5367 for another way of considering the relationship between PL, PA and health outcomes.  Additionally, there was no measure of PA for the present sample – they may not meet criteria for EDD, so how could data from these children be useful when considering EDD?

By having a focus on PA and EDD in the introduction, there is a lack of explanation, conceptualisation and evidence reporting on PL.  The definition used for PL is from a Canadian Assessment of Physical Literacy (CAPL) rather than the revised version (CAPL-2).  Additionally, the introduction lacks detail of the broader international literature and current conceptualisations of PL - see Cairney, John, et al. "Physical literacy, physical activity and health: Toward an evidence-informed conceptual model." Sports Medicine 49.3 (2019): 371-383. 

Author Response

General Comments: The purpose of the study was to evaluate tests of physical literacy and to examine sex differences.  I couldn’t see any evidence of test evaluation in this manuscript.  I also couldn’t see any evidence of controlling for chance findings that are likely to occur with any categorisation – in this case sex (And why not gender? It wasn’t clear how this information was obtained).  The participants were recruited from YMCAs and the sample size was quite small, so it is difficult to claim these participants are representative.  I can see this is part of a larger study and perhaps more data from the larger study need to be included to make this manuscript more meaningful.

Authors’ Response: We appreciate your insightful comments.

For the categorisation, we decided to ask a biological sex instead of gender because we believed that the participants are too young to accurately establish their gender identify.

We admit that our sample size is relatively small. Your point was added in our limitation.

The current study focused on evaluating physical literacy through various physical tests, and potential sex effects. We had a set of questionnaires in the current study. However, another author is currently synthesizing both physical tests and the questionnaire.   

Comment 1. The design flaws and the overall lack of significance of the research is difficult to overcome with revisions.  In case revisions are requested, I have included the following comments.

Response 1: Thank you for your comments. We descriptively assessed available physical test data and compared by sex. This is the focus of the current study, and I hope some readers will find values in the current study. 

Comment 2. Abstract – Please make sure it complies with MDPI requirements of 200 words or less – see 2.4 https://www.mdpi.com/authors/layout

Response 2: Thank you for your suggestion. We formatted our abstract under 200 words. 

Comment 3. Full stop/period should be after the citation, not before.

Response 3: Thank you for your words. We switched the order of citations and periods in the whole manuscript.

Comment 4. The first paragraph has a focus on physical activity (PA) rather than physical literacy (PL) and there is reference to exercise deficit disorder (EDD) throughout the introduction without establishing a clear connection between EDD and PL – I also don’t think it is necessary to make this case. That said, if the authors want to make the case that PL is important because it could lead to strategies to overcome EDD (line 76), then they need to explain how PL relates to PA (and how PA relates to EDD).  These are not direct causal relationships.  Please also see Caldwell, Hilary AT, et al. "Physical literacy, physical activity, and health indicators in school-age children." International Journal of Environmental Research and Public Health 17.15 (2020): 5367 for another way of considering the relationship between PL, PA and health outcomes.  Additionally, there was no measure of PA for the present sample – they may not meet criteria for EDD, so how could data from these children be useful when considering EDD?

Response 4: Thank you for your suggestions, especially the key literature. Based on your input, we restructured our introduction. Hopefully, each aspect is more inter-related and give sufficient background in the introduction.

Also, about the PA, another author in this study is currently working on another manuscript based on a connection between physical tests and questionnaire.  For this paper, only available data are physical tests.

Comment 5. By having a focus on PA and EDD in the introduction, there is a lack of explanation, conceptualisation and evidence reporting on PL.  The definition used for PL is from a Canadian Assessment of Physical Literacy (CAPL) rather than the revised version (CAPL-2).  Additionally, the introduction lacks detail of the broader international literature and current conceptualisations of PL - see Cairney, John, et al. "Physical literacy, physical activity and health: Toward an evidence-informed conceptual model." Sports Medicine 49.3 (2019): 371-383. 

Response 5: Thank you for your recommendation and giving an additional literature.  We made a major revision in our introduction based on your guidance along with the suggested literature.

Round 2

Reviewer 2 Report

Thank you for the extensive revisions and explanation of how these data fit within a larger project.

Despite the revisions, the scope of the study seems limited and descriptive.  I see that the plan is to publish the PA data elsewhere, but wondered if the PA data could also be incorporated into this manuscript in a way that is unique and doesn’t breach any originality requirements for other publications.  Otherwise, perhaps the PA data can be incorporated at a later stage after the main PA data have been published.  There should be a way to do this with acknowledgement of previous publications.  It is essential to have enough PA data to know the number of participants who meet criteria for EDD given the claim that the findings have implications to overcome EDD. A revision needs to include PA data.

In the previous review, I mentioned that I couldn’t see how the PL tests had been evaluated.  An evaluation usually means obtaining measures of reliability, validity or other forms of analysis such as cost-benefit.  All I can see is that the tests were used.  I’m not able to see evidence of evaluation.  There are differences in the author responses and the amended text in the manuscript.  The author response refers to “evaluation physical literacy”  while the amended text (line 21) as well as original text (line 81) states the purpose is to “evaluate selected tests of physical literacy”.  I can see that there was an evaluation of participants’ physical literacy, but not an evaluation of selected tests.  If the latter was available, this manuscript could make an interesting contribution.

Author Response

General Comments: Thank you for the extensive revisions and explanation of how these data fit within a larger project.

Authors’ Response: Thank you for your valuable comments and suggestions in the previous round.  We feel your input strengthened our manuscript.

Comment 1. Despite the revisions, the scope of the study seems limited and descriptive.  I see that the plan is to publish the PA data elsewhere, but wondered if the PA data could also be incorporated into this manuscript in a way that is unique and doesn’t breach any originality requirements for other publications.  Otherwise, perhaps the PA data can be incorporated at a later stage after the main PA data have been published.  There should be a way to do this with acknowledgement of previous publications.  It is essential to have enough PA data to know the number of participants who meet criteria for EDD given the claim that the findings have implications to overcome EDD. A revision needs to include PA data.

Response 1: Thank you for your comments. We realized how PA data are crucial in this manuscript. We added “physical activity levels” as a sub-heading in result section and added MVPA related information in the section (line 279-285).

Also, since we did not have PA information from all tested participants, we added it in the limitation section (line 364-365). 

Comment 2. In the previous review, I mentioned that I couldn’t see how the PL tests had been evaluated.  An evaluation usually means obtaining measures of reliability, validity or other forms of analysis such as cost-benefit.  All I can see is that the tests were used.  I’m not able to see evidence of evaluation.  There are differences in the author responses and the amended text in the manuscript.  The author response refers to “evaluation physical literacy”  while the amended text (line 21) as well as original text (line 81) states the purpose is to “evaluate selected tests of physical literacy”.  I can see that there was an evaluation of participants’ physical literacy, but not an evaluation of selected tests.  If the latter was available, this manuscript could make an interesting contribution.

Response 2: Thank you for your words. We understood your point. Instead of “selective tests of physical literacy,” we changed the expression to “selected physical tests.” This change was made at line 20, 80, 86, and 313.  Also, we removed “physical literacy” from our title.